# Reasons for Migration, Post-Migration Sociocultural Characteristics, and Parenting Styles of Chinese American Immigrant Families

**DOI:** 10.3390/children10040612

**Published:** 2023-03-24

**Authors:** Xinyue Wang, Stephanie L. Haft, Qing Zhou

**Affiliations:** Department of Psychology, University of California, Berkeley, CA 94720, USA

**Keywords:** immigrant, Chinese, migration, cultural orientations, socioeconomic status

## Abstract

With the growing percentage of Chinese immigrants in the U.S. population, it is crucial to understand how pre-migration factors (such as reasons for migration) are related to the adjustment of families in the host country. The present study examined reasons for migration and their associations with post-migration sociocultural factors and parenting styles in a community-based sample of Chinese American immigrant families (N = 258) living in the San Francisco Bay Area. The parents’ self-reported reasons for migration included family reasons (55.1%, e.g., family reunification), betterment reasons (18.0%, e.g., better education and occupational opportunities), and both family and betterment reasons (26.9%). Those who migrated for betterment reasons reported significantly higher parental education and per capita income than the family reason group (*p* < 0.001) and significantly higher income than the combined reason group (*p* = 0.007). No significant group differences emerged in cultural orientations and parenting styles after controlling for socioeconomic factors. The findings suggested that Chinese immigrant families who migrated solely for better education and occupational opportunities had significantly higher post-migration socioeconomic status than other reason groups. These differences have relevance for programs and services for new immigrants, as families might need different types of support (e.g., socioeconomic vs. relational) depending on their motivations for migration and post-migration socioeconomic resources.

## 1. Introduction

China is one of the top sending countries of immigrants to the U.S. [1]. In addition to adults and children who migrated from Mainland China, Hong Kong, Macau, and Taiwan (first-generation immigrants), there is a growing number of children born in the U.S. with one or two foreign-born parents (second-generation immigrants) [2]. With the increase in the U.S. immigrant populations, there is a high need to better understand the heterogeneity in the adjustment of immigrant families. While existing research has examined the links of post-migration sociocultural factors (e.g., acculturation, socioeconomic status) to immigrant families’ functioning and child adjustment [3,4,5], few researchers have examined pre-migration factors (such as reasons for migration) and their links to post-migration family functioning.

According to segmented assimilation theory, the societal adjustment of immigrants in the host country depends on the interaction of pre-migration exit factors, as well as the post-migration reception context [6]. Pre-migration factors include the reasons for migrating and the related sociocultural resources that immigrants bring with them to the host country (financial, skills, knowledge) [7]. These factors may have reverberating influences on the next generation, as immigrant parents may strategically adopt certain parenting styles to facilitate their goals for themselves and their children in the host country [8]. Research on migration reasons has historically focused on immigrants broadly—however, studies of Chinese immigrants specifically are needed given their growing numbers and the unique circumstances in their home country. Therefore, the present study aimed to characterize the reasons for migration and examine their relation to post-migration sociocultural characteristics and parenting styles in a sample of Chinese American immigrant parents with school-aged children. Understanding the link between reasons for migration and immigrants’ adaptation in the host country has implications for both immigration policy and support services for new immigrants.

### 1.1. Reasons for Migration among Chinese American Immigrant Families

Previous research has shown that there can be a variety of reasons for migration/immigration, including economic reasons, family reunification, political issues, and natural disasters [9]. Few researchers have investigated the reasons for migration among Chinese immigrants specifically. An exception is Lobo and Salvo (1998) [10], who examined U.S. immigration data and found that there are three main reasons for Asians to migrate to the U.S.—employment, seeking political refuge, and joining family members. Reasons for migration could be driven by pre-migration “push factors” pertaining to conditions in the country of origin, as well as “pull factors” involving immigration policies in the host country. In terms of country-of-origin context, Chinese immigrants’ reasons for migrating may be influenced by the economic growth of immigrants’ home countries/regions. This growth has cultivated a group of people who lived in urban areas, were more educated and skilled, and had more socioeconomic resources to support them in seeking educational and employment opportunities abroad [11]. Unlike the 19th and early 20th centuries when Chinese immigrants came to the U.S. primarily as laborers, nowadays, educational and employment opportunities have been endorsed as primary reasons for migration in studies of Chinese immigrants [12]. China has been the primary source of international students enrolled in higher education in the U.S. [2]. Fewer than 5000 Chinese people migrated to the U.S. as temporary workers in 1996 [13], but more than 60,000 were admitted in 2019 [14]. In summary, driven by the changing economic circumstances in Mainland China and adjacent regions, there is likely an increasing number of Chinese immigrants who have migrated to the U.S. mainly for voluntary choices in recent decades, especially to pursue education and employment.

The change in the U.S. immigration system and the enactment of the 1965 Immigrant and Nationality Act set the foundation for family reunification and skill-based immigration for Chinese individuals [15]. Currently, U.S. immigrant visas are grouped into two main categories: work visas sponsored by employers and family reunion visas sponsored by family members in the host country. The number of employment-based visas is limited each year [16]. Therefore, people who do not have immediate relatives and families in the U.S. alternatively enter the U.S. under nonimmigrant visa status instead. According to recent estimates, approximately 30% of Chinese individuals who became permanent residents in the U.S. did so through employment-sponsored routes, while the remainder qualified through immediate relative and family-sponsored visas [2]. Therefore, the current U.S. visa policy has rendered it more likely for Chinese immigrants to migrate to the U.S. for family reasons and have family members already living in the U.S.

### 1.2. Relations between Reasons for Migration and Post-Migration Sociocultural Characteristics

Chinese immigrant families’ reasons for migration may be associated with differences in their post-migration socioeconomic status and cultural orientations in the host country. According to the healthy immigrant hypothesis, immigrants tend to positively self-select before migration [17,18], meaning that those in better health and with more resources make the migration decision more willingly. Compared to the average foreign-born population in the U.S., immigrants from China are more likely to have higher levels of education [2], which may be associated with employment opportunities with higher wages and better benefits [19]. Immigrants who migrate for educational reasons may accumulate their socioeconomic capital as they improve their knowledge and skills, facilitating the goal of attaining jobs with higher incomes. Over the past few decades, a highly educated and skilled immigrant middle class has emerged from the rise in Asian migration, with education viewed as a pathway to achieve social mobility [20]. This upward mobility may be more accessible for individuals who migrated at a younger age—the chances of socioeconomic success were found to be 102% higher in individuals who immigrated before age 35 [21]. Studies have revealed that females who migrate for family reasons may have a lower income due to family obligations that could compromise their job and income outcomes [22]. Taken together, studies of Chinese immigrants showed that migrating for educational reasons may be associated with higher post-migration socioeconomic status, whereas migrating for family or political reasons may be related to lower post-migration socioeconomic status.

Cultural orientation is a bi-dimensional psychological construct referring to the extent to which an immigrant engages in the traditions, norms, and practices of the host and heritage cultures [23]. Previous research has suggested that the cultural orientation of Chinese immigrants can be influenced by the communities in which immigrants reside [24]. The Chinese immigrants who migrated to join family members may settle in ethnic enclaves (e.g., Chinatown), which often provide ample resources for heritage culture socialization [2]. In a study of Chinese American adolescents, residing in more ethnically dense neighborhoods was associated with a stronger Chinese cultural orientation and a weaker American orientation [25]. In addition, residing in intergenerational households is relatively common in Chinese immigrant families, and studies suggest that grandparents play active roles in passing down Chinese traditions to their grandchildren [26]. Conversely, if immigrants migrate to achieve better socioeconomic circumstances, they may be more motivated to attain (or may already have) higher English language proficiency to acculturate to American culture. A study of Chinese immigrant working women showed that low English proficiency was a barrier to achieving higher occupational and financial status in the U.S. [27]. Chinese immigrant parents who seek better education for their children may view immersion in American culture as a necessary pathway for later career success [28]. Overall, migrating for family-based reasons may be associated with a higher Chinese cultural orientation, whereas migrating for socioeconomic betterment may be associated with a higher American cultural orientation. However, these hypotheses have not been empirically tested in Chinese immigrant families.

### 1.3. Relations between Reasons for Migration and Parenting Styles of Immigrant Parents

The development and adjustment of first- and second-generation immigrant children are greatly influenced by parenting practices [29]. Among the characterizations of parenting practices, the styles of authoritative and authoritarian parenting differ in the amount of nurturing a child receives and the extent to which a child’s activities and behaviors are structured or controlled [30]. Specifically, authoritative parents tend to have high parental warmth and acceptance and adopt praise, inductive reasoning, and non-punitive disciplinary practices [31]. By contrast, the authoritarian parenting style is characterized by low parental warmth and acceptance, high parental control, and frequent use of punitive disciplinary practices. According to a meta-analytic study of 428 studies sampling families of diverse cultural backgrounds, authoritative parenting has been generally associated with positive developmental outcomes in children and adolescents (*r*s = 0.14–0.18), and authoritarian parenting has been generally associated with more internalizing and externalizing problems and poorer academic performance (*r*s = 0.09–0.16) [32]. However, most of the studies were conducted in Western countries or urban areas, so parental behaviors may not generalize beyond these contexts.

Immigrant parents’ parenting practices are often reconstructed during migration [33]. Cross-cultural studies found that compared to European American parents, Chinese parents tended to endorse higher authoritarian parenting and lower authoritative parenting [34,35]. Families who migrate to the U.S. may have different expectations of their future life and the outcomes of their children depending on the reasons for migration, which could influence parents’ practices in educating their children [36]. When encountering immigration-related stressors such as economic challenges and social instability, parents may seek to provide security and a sense of control for their children with limited resources, thus, increasing levels of authoritarian parenting and decreasing levels of authoritative parenting [35]. From a Chinese cultural perspective, authoritarian parenting that emphasizes conformity to parental expectations may facilitate Chinese parents’ goal for their children to be more hardworking and self-disciplined [37]. A study of Chinese immigrants in the Netherlands found that individuals migrating from rural areas of China and with fewer socioeconomic resources tended to adopt more authoritarian parenting styles, potentially as a way of maintaining traditional Chinese values [38]. In contrast, Chinese immigrants who migrated from urban areas of China and sought more skilled occupations in the host country adopted more authoritative parenting styles post-migration. There is heterogeneity in parenting styles among Chinese American immigrants. Studies have found that mothers who have a higher level of American cultural orientation or integrated participation in both American and Chinese cultures are likely to adopt more authoritative parenting and less authoritarian parenting [39]. Across cultures, lower socioeconomic status tends to be associated with lower authoritative parenting and higher authoritarian parenting, partially due to higher family stress associated with socioeconomic disadvantages [40,41]. Taken together, these studies suggest that migrating for socioeconomic betterment and with more socioeconomic resources may result in more authoritative parenting in the host country, while migrating for family reasons may be associated with more authoritarian parenting.

### 1.4. The Receiving Context of the San Francisco Bay Area

Segmented assimilation theory posits that one key factor of post-migration receiving contexts influencing immigrant adjustment is the strength and viability of the ethnic and immigrant community [7]. In the U.S., data show that almost half of the Chinese immigrants reside in three major metropolitan areas: New York City, San Francisco, and Los Angeles. These areas contain historical ethnic enclaves (“Chinatowns”) that may provide immigrants with more social capital and support in the U.S. However, more recently, immigrants have been settling in states such as North Carolina, Georgia, and Nevada due to expanding job markets in technology sectors [42]. The present study sampled the San Francisco Bay Area, which represents a geographic location that is a combination of an expanding technology industry and a high concentration of ethnic Chinese communities. The San Francisco Bay Area has a long history of Chinese immigration, with heterogeneity in socioeconomic and occupational experiences. This receiving context, therefore, is a unique and ideal location in which to study variability in reasons for migration and sociocultural characteristics.

### 1.5. Present Study

The present study used the first wave (Wave 1) of data from a longitudinal study of a community-based sample of Chinese immigrant families in the San Francisco Bay Area (*N* = 258; masked for blind review). The overall goal of the study is to characterize the families’ reasons for migration and how these reasons relate to their post-migration sociocultural factors and parenting styles. Specifically, the first aim of the study is to descriptively characterize the families’ self-reported reasons for migration. Based on prior research and current visa immigration policies [15], we hypothesized that the most common reason for migrating would be family-based (e.g., family reunification). The second aim of the study is to investigate the associations between sociocultural characteristics (socioeconomic status and cultural orientations) and families’ reasons for migration. We hypothesized that families who migrated for socioeconomic betterment would have higher parental education and family income and that parents would be more oriented toward the American culture compared to families who migrated for other reasons. In contrast, we hypothesized that families who migrated for family-related reasons would have lower parental education and family income, and the parents would have higher Chinese culture orientation. The third aim of the study is to explore how reasons for migrating are associated with parenting styles. Based on a prior study of Chinese immigrants in the Netherlands [38], we hypothesized that those who migrated to seek socioeconomic betterment might report higher authoritative parenting compared to the families who migrated for other reasons.

## 2. Materials and Methods

### 2.1. Participants

The present study used Wave 1 data from a longitudinal study of a community-based sample of 258 Chinese American immigrant families with school-age children (masked for blind review). To be eligible for the study, participant families had to meet the following eligibility criteria at the time of Wave 1 screening: (a) the child was in either the first or second grade, (b) the child was either a first- (born out of the U.S.) or second-generation (born in the U.S. with at least one foreign-born parent), (c) the child was living with at least one of their biological parents for the most of the time, (d) both the child and the parent understood and spoke English or Chinese, and (e) both parents identified themselves as Chinese ethnicity.

Of 258 children who participated in Wave 1 data collection (conducted between December 2007 and July 2009), 61 were first-generation (23.64%), and 197 were second-generation (76.36%). Approximately half of the children were boys (51.94%). The children were between the ages of 5.81 and 9.14 (mean age = 7.38). At the time of the interview, most children were in either the first grade (48.84%) or the second (50.0%). In the sample of adult parents, 211 were the mother of the participating child (81.78%), and 47 were the father (18.22%). After omitting parents who did not report reasons for migration, 245 participants were included in the data analysis. The descriptive statistics of the study variables are shown in Table 1. Compared with the 245 participants who specified reasons for migration, those who did not (*N* = 13) were significantly older [*t*(249) = −3.0821, *p* = 0.002] and had higher annual family per capita income [*t*(246) = −2.0311, *p* = 0.043], but did not differ in age of immigration, years in the U.S., education level, birth country, or employment status. Among the 245 parents who reported their reasons for migration, the majority (n = 243, 99.18%) were foreign-born, including mainland China (n = 187, 76.33%), Hong Kong (n = 23, 9.39%), Taiwan (n = 8, 3.27%), or other parts of the world (n = 25, 10.20%). Most parents were in their thirties and forties (*M* = 39.26 years, range = 27.88 to 53.94 years, *SD* = 5.13 years). On average, parents had lived in the United States for more than a decade (*M* = 11.75 years, range = 0.5 to 38 years, *SD* = 7.49 years). Parents’ level of education ranged from elementary school education (5 years) to a doctoral degree or other advanced degrees (20 years), with a mean level of 13.26 years of education (*SD* = 2.47 years). Families’ per capita income ranged from USD 625 to USD 50,000 (*M* = USD 11,390.39, range = USD 625 to USD 50,000, *SD* = USD 8196.42). Of the parents who reported their employment status, 136 (64.15%) were employed full-time, 26 (12.26%) were employed part-time, 7 (3.30%) were occasionally employed or as day-labor, and 43 (20.28%) were unemployed or homemakers.

### 2.2. Procedures

Participants were recruited from Chinese American immigrant families living in the San Francisco Bay Area. Researchers recruited participants through recruitment fairs conducted in Asian or Chinese American communities (e.g., grocery stores, shopping centers, and neighborhood events), partnerships with schools, referrals from community organizations, and distributing fliers. Participants were screened by trained bilingual research assistants through phone calls to determine eligibility in their preferred language. Participants then gave informed consent after the phone screening. Ultimately 258 families consented to take part in and completed the Wave 1 assessment. A total of 258 children completed a 2.5-hour laboratory assessment with one parent. All interviews and tests were administered in the parent’s or child’s preferred language (English, Mandarin, or Cantonese). All written materials, including consent and assent forms and questionnaires, were available in English, Simplified Chinese, or Traditional Chinese. Toward the end of the laboratory visit, parents received compensation of USD 50 and reimbursement for transportation, and children were given a small prize. The present study used data collected from the parent interview and questionnaires, specifically, the family demographics and parenting styles questionnaires. All procedures were approved by the Institutional Review Board of (masked for blind review).

### 2.3. Measures

Family Demographics and Migration History. All questionnaires used in the present study are available in Appendix A. An adapted and translated version of the Family Demographics and Migration History Questionnaire [43] was administered in the parent interview to obtain demographic information and reasons for migrating. The basic demographic information used in the present study included date of birth, age of immigration, country of origin, parental education, and per capita income. Items related to ethnicity and country of origin were modified to suit Chinese immigrant families. There were 9 reasons for moving to the U.S. listed for families to select from, (1) to join family members; (2) to leave political problems; (3) to leave personal problems; (4) to find a good job or earn a better income; (5) because your family brought you; (6) to get an education for yourself; (7) to get married; (8) to provide your children with an education or better opportunities; and (9) other reasons. The participating parent answered yes or no to each reason on behalf of their family according to their situations, with the option to select multiple reasons. Those who chose item 9 were asked to specify the reason other than the listed.

Parenting Styles. Parents rated their own parenting styles by using the Parenting Styles and Dimensions Questionnaire (PSDQ, [44]). The Chinese version of the PSDQ had satisfactory internal reliabilities in previous research with Chinese populations [45,46]. For each item, parents used a 5-point Likert scale to rate how often they exhibit the behavior with their children (from 1 = Never to 5 = Always). Composite scores of authoritative and authoritarian parenting styles were formed by averaging the corresponding item scores. The alpha reliabilities were 0.90 for authoritative parenting (17 items) and 0.78 for authoritarian parenting (13 items).

Parent American and Chinese Cultural Orientations. The Cultural and Social Acculturation Scale (CSAS, [47,48]) was used for parents to report their own orientations towards Chinese and American cultures. The CSAS is a bi-dimensional scale (32 items) which assesses individuals’ contact with and engagement in both heritage and host cultures. Both Chinese and English versions of the CSAS have shown satisfactory internal reliability [49]. The CSAS assesses parents’ bi-dimensional cultural orientations primarily in three domains: (1) language proficiency (four items for Chinese proficiency and four items for English proficiency), (2) media use (five items on Chinese media use and five items on English media use), and (3) social relationships or friends (three items on Chinese friends and three items on Caucasian-American friends). Items were rated on Likert scales ranging from 1 to 6 points. We computed the composite scores of parents’ overall Chinese and American orientations by averaging the corresponding item scores. Standardized values were then computed from the composite scores. The alpha reliabilities in this sample are 0.73 and 0.87 for parents’ Chinese cultural and American cultural orientations, respectively.

### 2.4. Analytic Plan

All statistical analyses were conducted in R version 4.1.1 [50]. To characterize reasons for migration (Aim 1), we computed the relative frequency of each reason. To investigate Aim 2 (relations with sociocultural factors) and Aim 3 (relations with parenting styles), we conducted a one-way ANOVA to test for differences in sociocultural factors (parents’ age of immigration, parents’ birth country and region, parental education, and family per capita income), cultural orientations, and parenting styles between parents who migrated for different reasons. For the analysis of variance in cultural orientations and parenting styles, we controlled for parental education and per capita income to eliminate any effects caused by socioeconomic factors using one-way ANCOVAs. Significant ANCOVAs and ANOVAs were followed up by post-hoc analyses. We used the Benjamini–Hochberg procedure [51] to correct p values for multiple comparisons.

## 3. Results

### 3.1. Aim 1: Characterizing Families’ Reasons for Migration

According to frequency analysis (Figure 1), the families’ listed reasons for migration, in order of frequency, were to join family members (118, 27.44%), to provide your children with an education or better opportunities (79, 18.37%), because your family brought you (77, 17.91%), to find a good job or earn a better income (48, 11.16%), to get an education for yourself (47, 10.93%), to get married (47, 10.93%), to leave personal problems (3, 0.70%), and to leave political problems (3, 0.70%). The remaining 8 (1.86%) parents endorsed other reasons that were not listed. Based on prior literature and for ease of analysis, immigrant families were grouped into three categories based on reasons for migration: (a) those migrated only for family-based reasons (*N* = 135, 55.10%), which included “to join family members”, “because your family brought you”, and “to get married”; (b) those migrated only for betterment reasons (*N* = 44, 17.96%), which included “to provide your children with an education or better opportunities”, “to find a good job or earn a better income”, “to get an education for yourself”, “to leave personal problems”, and “to leave political problems”; and (c) those migrated for both family-based and betterment reasons (N = 66, 26.94%).

### 3.2. Aim 2: Examining the Relations between Reasons for Migration and Sociocultural Factors

Before comparing reasons for migration groups on sociocultural factors (age of immigration, birth country/region, education level, per capita income, and cultural orientations) using a one-way ANOVA, we checked that the assumptions of ANOVA were met (normality, sample independence, and variance equality). For variables that did not meet the homogeneity of variance (education level, per capita income, and authoritarian parenting), we used Welch’s ANOVA, followed by the Games–Howell post-hoc test.

As shown in Table 2, parents who migrated for betterment reasons had, on average, 14.6 years of education (SD = 3.04), while the family reason group had, on average, 12.7 years of education (SD = 2.21) and the multiple reason group had, on average, 13.4 years of education (SD = 2.18). A one-way Welch ANOVA showed that education was significantly different across different reason groups [F(2,97.4) = 8.23, *p* < 0.001]. The Games–Howell post-hoc analyses revealed that the differences in education between betterment and family (−1.92, 95% CI [−3.11, −0.72]) groups were statistically significant (*p* = 0.007). Differences in education between the betterment and multiple reasons groups and between the family and multiple reason groups were not statistically significant.

Differences across betterment, family, and combined betterment/family groups in per capita income were analyzed using a Welch one-way ANOVA since the homogeneity of variance was violated. Results demonstrated that per capita income was significantly different across three groups [F(2, 94) = 7.02, *p* = 0.0014]. Specifically, families that migrated for betterment reasons had an average per capita income of USD 16,414 (SD = USD 10,283). The family reason group had, on average, USD 10,077 in per capita income (SD = USD 7118), while those who migrated for multiple reasons had an average of USD 10,684 in per capita income (SD = USD 7499). The Games–Howell post-hoc analyses showed that the family reason group had a significantly lower per capita income (−6337, 95% CI [−10,403, −2272]) than the betterment reason group (*p* = 0.049). The multiple reasons group also had a significantly lower per capita income (−5730, 95% CI [−10,086, −1374]) than the betterment reason group (*p* = 0.049). There was no significant difference between the family and multiple reasons groups.

Significant group differences were found between migration groups in American cultural orientations [F(2, 188) = 3.88, *p* = 0.022] according to a one-way ANOVA. Families who migrated only for family reasons (M = −0.05, SD = 0.60) had a significantly lower American cultural orientation (−0.32), 95% CI [−0.582, −0.0478], *p* = 0.02) than those who migrated only for betterment reasons (M = 0.26, SD = 0.62). A one-way ANCOVA was employed to detect whether this difference remained significant after controlling for socioeconomic status (parental education and per capita income). American cultural orientation was not significantly different across different reason groups [F (2, 180) = 0.067, *p* = 0.94] after controlling for parental education and per capita income.

No significant group differences emerged between the three categories of migration reasons in Chinese cultural orientation, age of immigration, or birth country/region (all *p*s > 0.05).

### 3.3. Aim 3: Relations between Reasons for Migration and Parenting Styles

Before comparing reasons for migration groups and parenting styles, we confirmed that the assumptions of ANOVA were met (normality, sample independence, and variance equality). A one-way ANOVA indicated that group differences in authoritative parenting were not significant (F (2, 239) = 0.50, *p* = 0.61). A one-way Welch ANOVA test showed significant group differences in authoritarian parenting [F(2, 83.72) = 3.33, *p* = 0.041], although the post-hoc test showed no statistically significant pairwise differences. We used ANCOVA to test whether group differences remained significant after controlling for education level and per capita income. Results from a one-way Welch’s testing for group differences in authoritarian styles controlling for education level and per capita income showed that differences in authoritarian parenting styles across groups were significant [F (2, 203) = 4.91, *p* = 0.008]. The betterment group had an average authoritarian parenting style of 2.19 (SD = 0.43), the family group had an average value of 2.08 (SD = 0.36), and the multiple groups had an average of 2.24 (SD = 0.48). However, a Games–Howell test did not reveal any significant pairwise differences.

## 4. Discussion

In this community-based sample of Chinese American immigrant families, most families reported migrating for family-related reasons, which aligns with previous research on U.S. immigrants and immigration policies [10,12]. The most commonly cited migration reason was to join family members, while the least common reason was to leave personal or political problems. Parents who migrated only for betterment reasons (to improve economic or educational circumstances) had significantly higher education levels than those who migrated only for family reasons and significantly higher per capita income than those who migrated for only family or both betterment and family reasons. Overall, individuals who did not migrate for family reasons had a significantly higher socioeconomic status, echoing previous studies [19,22]. Parents who only migrated only for betterment reasons had significantly higher American cultural orientation than those who only migrated for family-related reasons, although this difference was not significant after controlling for parents’ education level and family per capita income. No significant differences between the three reason groups were found in authoritative or authoritarian parenting.

### 4.1. Reasons for Migration

According to the frequency distribution of eight different reasons, the most common reason parents migrated to the U.S. was to join family members, which was consistent with the initial hypothesis. The second most common reason was to provide children with education or better opportunities. The least common reason was to leave personal or political problems. After categorizing families into three mutually exclusive groups, most parents migrated solely for family reasons, including joining family members, getting married, and being brought by the family, while the fewest parents cited seeking socioeconomic opportunities and improving circumstances as their sole reason for migration. The findings suggest that the migration to the U.S. during the 1980s and 2000s was mainly motivated by family-related matters.

The reasons and motivations behind immigration are often related to the immigration policy of the host country [12]. Family-sponsored visas are more available than employment-sponsored visas for people to permanently live and work in the United States [52], which explains why most Chinese immigrants likely obtained their legal status through family members. In the present sample, participants migrated to the U.S. between the 1980s and the early 2000s, an era during which China started a series of economic reforms and rose to be one of the world’s largest economies [53]. In the 1980s and 1990s, most Chinese families did not have the financial capability to support their children or themselves as international students. However, with economic growth in China, it has become more common for Chinese immigrants to enter the U.S. as international college students [2]. Parents endorsed providing children with education or better opportunities as the second most common reason for immigration, which is in line with the Chinese tradition of valuing education [54,55].

### 4.2. Reasons for Migration and Sociocultural Characteristics

The second aim of the study was to examine whether reasons for migration were associated with the sociocultural characteristics (age of immigration, birth country, parental education, per capita income, Chinese and American cultural orientations) of Chinese immigrant families. Results showed that reasons for migration did not relate to the age of immigration and birth country and region. However, our findings were consistent with our hypothesis that immigrants who migrated for betterment purposes would have higher education and better income post-migration, but those who migrated for family-related reasons had lower post-migration socioeconomic status.

Families who migrated to improve their circumstances may have had higher socioeconomic capital and resources prior to their immigration, consistent with findings on the positive selection of immigrants [18]. Increasing their knowledge and skills in the host country may have then subsequently enhanced their education and income levels even further. However, some research suggests that Chinese immigrants face a “glass door” in their employment and earnings opportunities, whereby their foreign credentials are devalued in the host country’s labor market [56]. Future research that tracks the pathway from pre-migration socioeconomic status, migration reason, and post-migration socioeconomic status will be informative in understanding the economic mobility of Chinese immigrants. A potential explanation for the lower income in those who migrated for family-based reasons is that when immigrants migrate to a new country for family reasons, they may make a compromise between personal career development and family cohesion [22].

Although we hypothesized that immigrants who migrated for family reasons would have a higher Chinese cultural orientation, we found no significant difference between the three groups in Chinese and American cultural orientations. Of note, our measure of cultural orientations included only behavioral acculturation—however, acculturation also involves changes in identity and values, which may have been associated with reasons for migration [57]. Perhaps reasons for migration would be more tightly linked to cultural orientations in the years just after migrating. However, in the present sample, participants had resided in the U.S. for 12 years on average, so other, more proximal factors encountered in the host country (e.g., intercultural contact, discrimination, ethnic community) may be more tightly linked to cultural orientations than the original reason for migration. Specifically, all participants were parents of young children, so intercultural contact through parents’ involvement in children’s schooling may be a more salient influence on parents’ cultural orientations.

### 4.3. Reasons for Migration and Parenting Styles

No significant differences in authoritative parenting style were found between the three migration reason groups after controlling for socioeconomic status (parent education and family income). Although overall group differences in authoritarian parenting styles were different across the three groups, no significant pairwise differences emerged in post-hoc analysis—therefore, findings on differences in authoritarian parenting are inconclusive and require further replication. Perhaps reasons for migration tend to shape domain-specific parenting practices rather than global parent styles captured by measures of authoritative and authoritarian parenting. For example, a study of Chinese immigrant parents in the Netherlands found that migration histories were related to parent involvement in children’s schools [38].

### 4.4. Limitations, Implications, and Future Directions

The results should be interpreted by considering the study’s limitations. First, we do not have pre-migration data on immigrant parents. Therefore, we are unable to conclude whether post-migration sociocultural characteristics are the result of stability, upward mobility, or downward mobility after migrating. Second, because the sample was recruited from an urban-to-suburban area with a high concentration of Chinese immigrants, the finding might not generalize to other Chinese immigrant families living in other parts of the country. Third, data for the present study were collected between 2007 and 2009, which may not reflect current immigration circumstances (e.g., the COVID-19 pandemic). Fourth, we only examined authoritative and authoritarian parenting style dimensions—other dimensions, such as permissive parenting, may be a salient parenting style for Chinese immigrant families. Finally, the present study did not investigate variability in families’ post-migration experiences (e.g., discrimination, job support) in tandem with reasons for migrating. Future research may also consider investigating reasons for migration in newer Chinese immigrant destinations in the U.S., as well as longitudinally examining how pre-migration factors percolate through the adjustment of the second generation of Chinese immigrants.

Within our sample, located in a relatively ethnically dense geographic area of the U.S., the majority (82%) of Chinese immigrants endorsed family as one of or the only reason for migrating. This finding has implications for school-based programs that serve immigrant families with young children, which typically focus on parent outreach and promoting parent involvement. Our results suggest that school-based programs for immigrant families may consider leveraging these networks in targeting outreach to grandparents, aunts, uncles, and other family members in addition to parents.

Our findings suggest that reasons for migration may be related to variability in the level of support that families have in the host country. The families solely seeking betterment opportunities may lack a culturally protective family network in the host country and may benefit from programs that boost relational support (e.g., by offering social activities or groups with other Chinese immigrants). Parents seeking betterment opportunities without family members already in the U.S. may also require more childrearing and caregiving support from schools. In contrast, Chinese immigrants who migrate for family reasons may already have a larger caregiving network but may require more socioeconomic and occupational support (e.g., job search agencies and food assistance). Ultimately, programs that serve immigrant families should consider tailoring the type of support they provide based on families’ pre-migration motivations and post-migration resources.

## 5. Conclusions

In our study of Chinese American immigrant families in an ethnically dense metropolitan area, families reported migrating for family-based reasons (e.g., to join family members), betterment reasons (e.g., economic or occupational betterment), or both. The majority of families in our sample reported migrating for family reasons—these families reported lower socioeconomic status in terms of income and education compared to other reason groups. We found no difference in parenting styles between families who migrated for different reasons. Although caution is warranted when generalizing these findings to other immigrant groups and geographic regions, results broadly suggest that immigrants migrating for different reasons may require different types of support in the host country.

## Figures and Tables

**Figure 1 children-10-00612-f001:**
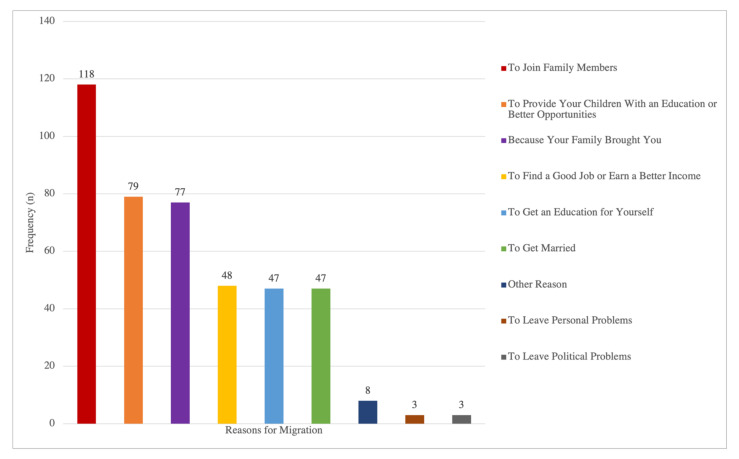
Frequency Distribution of Reasons for Migration.

**Table 1 children-10-00612-t001:** Descriptive Statistics of Study Variables.

Variables	Min	Max	Mean	SD	Skewness	Kurtosis
Demographics and Covariates						
Parent Age (years)	27.88	53.94	39.26	5.13	0.38	−0.38
Parents’ years in the U.S.	0.5	38	11.75	7.49	0.8	0.15
Parents’ age of immigration	1.35	46.51	27.36	7.7	−0.32	0.38
Socioeconomic Status						
Parent Education Level (years)	5	20	13.26	2.47	0.51	0.29
Annual per capita income (USD)	625	50,000	11,390.39	8196.42	1.4	2.27
Parent Cultural Orientation						
Chinese Cultural Orientation	−2.05	1.08	−0.01	0.49	−0.79	1.94
American Cultural Orientation	−1.15	2.12	0.05	0.6	0.55	0.14
Parenting Styles						
Authoritarian Parenting Style	1.26	3.74	2.14	0.41	0.97	1.35
Authoritative Parenting Style	2	4.93	4.08	0.49	−0.61	1.29

Note. *N* = 245.

**Table 2 children-10-00612-t002:** Descriptive Statistics of Study Variables by Reasons for Migration.

	Betterment Reason (*N* = 44)	Family Reason (*N* = 135)	Multiple Reason (Betterment + Family; *N* = 66)	Significant Differences
Age of Immigration (years)	27.7 (6.90)	27.2 (7.81)	27.4 (8.08)	*ns*
Parental Education (years)	14.6 (3.04)	12.7 (2.21)	13.4 (2.18)	betterment > family (*p* = 0.007)
Per Capita Income (USD)	16,414 (10,283)	10,077 (7118)	10,684 (7499)	betterment > family (*p* = 0.049) betterment > multiple (*p* = 0.049)
Chinese Cultural Orientation	0.023 (0.43)	−0.024 (0.54)	0.004 (0.43)	*ns*
American Cultural Orientation	0.28 (0.62)	−0.039 (0.61)	0.046 (0.55)	*ns*
Authoritative Parenting Style	4.04 (0.47)	4.11 (0.50)	4.04 (0.48)	*ns*
Authoritarian Parenting Style	2.19 (0.43)	2.08 (0.36)	2.24 (0.48)	*ns*

Note. Mean is displayed with the standard deviation in parentheses; *ns* = not significant per group mean comparison testing; all *p* values are corrected for multiple comparisons using the Benjamini–Hochberg method.

## Data Availability

Data are not publicly available because we did not have consent in favor of public access from the subjects involved.

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
