# Peer review of "Reasons for Migration, Post-Migration Sociocultural Characteristics, and Parenting Styles of Chinese American Immigrant Families"

_children, 2023, doi:10.3390/children10040612_

Round 1
Reviewer 1 Report
The study is well structured and presented; there are two main factors lowering its perceived relevance:
1. as reported in the conclusions, the geographical scope of the study and the investigated sampe do not allow to generalize conclusions. Sample is concentrated in a specific area (S.F. Bay Area) and we don't know if same results would be replicated if the investigation would be performed in other industrial or rural districts; socio-economical background information about the investigated sample are not known;
2. again, as reported in the conclusions, data were collected about 15 years ago.
Author Response
The study is well structured and presented; there are two main factors lowering its perceived relevance:
- as reported in the conclusions, the geographical scope of the study and the investigated sampe do not allow to generalize conclusions. Sample is concentrated in a specific area (S.F. Bay Area) and we don't know if same results would be replicated if the investigation would be performed in other industrial or rural districts; socio-economical background information about the investigated sample are not known;
We agree with this limitation and acknowledge this in section 4.4 of our Discussion:
[Pg. 13]: “… because the sample was recruited from an urban-to-suburban area with a high concentration of Chinese immigrants, the finding might not generalize for other Chinese immigrant families living in other parts of the country.”
We also acknowledge the unique geographic context of the study location in section 1.4 of our Introduction, “The Receiving Context of the San Francisco Bay Area.” By providing in-depth information on the geographic location of the present study, we hope other researchers can evaluate the relevance of the findings to their respective geographic areas (e.g. in terms of relative ethnic density or urban vs. suburban). Our study may provide valuable information on the adjustment of Chinese American immigrant families in areas of similar neighborhood environment and socioeconomic development. The San Francisco Bay Area is one of three metropolitan areas that almost half of Chinese immigrants reside in – thus, although findings may not be relevant to all geographic contexts, they still may inform understanding of a large portion of Chinese immigrant communities.
- again, as reported in the conclusions, data were collected about 15 years ago.
We agree with this limitation and acknowledge this in section 4.4 of our Discussion:
[Pg. 13]: “Third, data for the present study were collected between 2007 and 2009, which may not reflect current immigration circumstances.”
We still believe our findings are helpful in contributing to an understanding of trends in reasons for migration – for example, researchers can evaluate and compare how contemporary reasons for migration given by Chinese immigrant communities differ from our study. In addition, there is currently no theoretical evidence to suspect that the relations between reason for migration and parenting styles would change over the past 15 years.
Reviewer 2 Report
very interesting paper, but it can be improved:
1. Add the wave1 as supplementary matherial and reference it
2. In table 2 add p values, not text.
3. Add as supplementary material all statistic tables with posthoc test with, I think, bonferroni correction
4. In statistic analysis add ANCOVA analysisis and explain distribution of sample, in order to support the use of parametric tests.
5. Youa have 258 participants, perhaps it would be good to perform a regression analysis instead ANOVA tests
Author Response
very interesting paper, but it can be improved:
- Add the wave1 as supplementary matherial and reference it
We are unsure what Reviewer 2 meant by “add the Wave 1 as supplementary material”. The present paper used Wave 1 data from a longitudinal study. All study variables (reasons for migration, post-migration socio-cultural characteristics, and parenting styles) were assessed at Wave 1. We are happy to provide other information if Reviewer 2 can clarify the suggestion.
- In table 2 add p values, not text.
We have added exact p values to the “Significant Differences” column of Table 2.
- Add as supplementary material all statistic tables with posthoc test with, I think, bonferroni correction
We appreciate the suggestion to correct for multiple comparisons. We have now done this, using the Benjamini-Hochberg procedure since it is recommended as having more statistical power than the Bonferroni correction (Glickman et al., 2014).
Glickman, M. E., Rao, S. R., & Schultz, M. R. (2014). False discovery rate control is a recommended alternative to Bonferroni-type adjustments in health studies. Journal of clinical epidemiology, 67(8), 850-857.
- In statistic analysis add ANCOVA analysisis and explain distribution of sample, in order to support the use of parametric tests.
We have added the use of ANCOVA in our section 2.4 “Analytic Plan”:
[Pg. 8]: “For the analysis of variance in cultural orientations and parenting styles, we controlled for parental education and per capita income to eliminate any effects caused by socioeconomic factors using one-way ANCOVAs.”
We explain the justification for the use of parametric or nonparametric tests in our section 3.2 on reporting results from Aim 2:
[Pg. 9]: “Before comparing reasons for migration groups on sociocultural factors (age of immigration, birth country/region, education level, per capita income, and cultural orientations) using a one-way ANOVA, we checked that the assumptions of ANOVA were met (normality, sample independence, and variance equality). For variables that did not meet the homogeneity of variance (education level, per capita income, and authoritarian parenting), we used Welch’s ANOVA followed by the Games-Howell post-hoc test.”
We have added a similar explanation supporting the use of parametric tests to section 3.3 on reporting results form Aim 3:
[Pg. 11]: “Before comparing reasons for migration groups on parenting styles, we confirmed that the assumptions of ANOVA were met (normality, sample independence, and variance equality).”
- You have 258 participants, perhaps it would be good to perform a regression analysis instead ANOVA tests
We have reproduced all analyses using regressions rather than ANOVA tests and have arrived at the same results in terms of directionality and significance. Therefore, to preserve space, we only report the results of the ANOVA tests in the manuscript.
Reviewer 3 Report
Review for Manuscript entitled “Reasons for Migration, Post-Migration Sociocultural Characteristics, and Parenting Styles of Chinese American Immigrant Families”
Thank you for providing me with the opportunity to review the manuscript entitled “ Reasons for Migration, Post-Migration Sociocultural Characteristics, and Parenting Styles of Chinese American Immigrant Families”.
Congratulations on your current publication.
1. Introduction:
The paragraph that appears in the introduction doesn´t make sense since it belongs to the template for the correct publication. Delete from line 28 to 36.
Congratulations, Introduction is written clearly however it is very long. It is necessary to synthesize everything in the introduction. Review and eliminate data that are not relevant to the objectives of the study.
The aim of the study in the introduction section is written clearly
2. Materials and Methods
Correct description of each dimension of the methods section.
3. Results
Congratulations, Results section is written clearly.
The great work done has been demonstrated.
The tables and figures are very informative
4. Discussion
The discussion section is correct but should present further justification. For example, the first paragraph of the discussion section should be discussed with other scientific publications.
Review the scientific literature and justify it more extensively.
5. Conclusions
It would be necessary to include a conclusion section summarizing the final results. These conclusions should be in accordance with the objectives of the study.
Author Response
- Introduction:
The paragraph that appears in the introduction doesn´t make sense since it belongs to the template for the correct publication. Delete from line 28 to 36.
We thank the reviewer for catching this oversight and have deleted the template from the introduction.
Congratulations, Introduction is written clearly however it is very long. It is necessary to synthesize everything in the introduction. Review and eliminate data that are not relevant to the objectives of the study.
The aim of the study in the introduction section is written clearly.
We have cut some wording from the introduction and reduced it by several lines.
- Materials and Methods
Correct description of each dimension of the methods section.
We thank the reviewer for their attention to the Materials and Methods.
- Results
Congratulations, Results section is written clearly.
The great work done has been demonstrated.
The tables and figures are very informative
We thank the reviewer for their attention to the Results.
- Discussion
The discussion section is correct but should present further justification. For example, the first paragraph of the discussion section should be discussed with other scientific publications.
Review the scientific literature and justify it more extensively.
We have added citations to highlight how our findings compare to other publications in the first paragraph of the Discussion:
[Pg. 11]: “In our sample of Chinese American immigrant families, most families reported migrating for family-related reasons, which aligns with previous research findings and immigration policies10,12.”
[Pg. 11]: “Overall, individuals who did not migrate for family reasons had a significantly higher socioeconomic status, echoing previous studies19,22.”
- Conclusions
It would be necessary to include a conclusion section summarizing the final results. These conclusions should be in accordance with the objectives of the study.
We have added a conclusion section (section 5.0).
Round 2
Reviewer 2 Report
1. I refer to the survey, add as supplementary material and reference it
2. well, it is not neccesary, ok
3. ok
4. ok
5. ok
Author Response
- We have added the survey to Supplementary Materials and acknowledge this in our measures section: [Pg 7]: "All questionnaires used in the present study are available in Supplementary Materials."
Reviewer 3 Report
Review for Manuscript entitled “REASONS FOR MIGRATION, POST-MIGRATION SOCIOCULTURAL CHARACTERISTICS, AND PARENTING STYLES OF CHINESE AMERICAN IMMIGRANT FAMILIES”.
Thank you for providing me with the opportunity to review the manuscript entitled “REASONS FOR MIGRATION, POST-MIGRATION SOCIOCULTURAL CHARACTERISTICS, AND PARENTING STYLES OF CHINESE AMERICAN IMMIGRANT FAMILIES”.
Congratulations for the great work done.
Thank you for having made the modifications that the reviewers have requested to the manuscript. It is to be appreciated that all the requested changes have been carried out.
In this regard, the following scientific paper is suitable for publication

Author Response
We thank the reviewer for the time spent in offering valuable comments on this manuscript.